# Use of Coronary CT Angiography to Predict Obstructive Lesions in Patients with Chest Pain without Enzyme and ST-Segment Elevation

**DOI:** 10.3390/jcm10225442

**Published:** 2021-11-21

**Authors:** June-sung Kim, Youn-Jung Kim, Yo Sep Shin, Shin Ahn, Won Young Kim

**Affiliations:** Department of Emergency Medicine, University of Ulsan College of Medicine, Asan Medical Center, Seoul 05505, Korea; jsmeet09@gmail.com (J.-s.K.); yjkim.em@gmail.com (Y.-J.K.); irrusters@gmail.com (Y.S.S.); ans1023@gmail.com (S.A.)

**Keywords:** chest pain, emergency department, acute coronary syndrome, coronary computed tomography angiography

## Abstract

It is challenging to rule out acute coronary syndrome among chest pain patients without both ST-segment elevation in electrocardiography and troponin elevation at emergency departments (ED). The purpose of this study was to develop a prediction model for rapidly determining the occurrence of significant stenosis in coronary computed tomography angiography (CCTA). Retrospective observational cohort study was conducted with 904 patients who had presented with chest pain without troponin elevation and ST-segment changes and underwent CCTA between January 2017 and December 2018. The primary endpoint was the presence of significant stenosis on CCTA, defined as narrowing above 70% diameter. The logistic regression model was used for development a new predictive model. One hundred and thirty-four patients (14.8%) were shown severe stenosis. The independent associated factors for significant stenosis were age ≥65 years, male, diabetes, history of acute coronary syndrome, and typical chest pain. Based these results, we developed a new prediction model. The area under the curve was 0.782 (95% confidence interval 0.742–0.822). Moreover, score of ≥5 was chosen as cut-off values with 86.6% sensitivity and 56.4% specificity. In conclusion, among chest pain patients without ST changes and troponin elevation, the new score will be helpful to identify potential candidate for CCTA such as patients with significant stenosis.

## 1. Introduction

Chest pain is the second most common cause of patient visits to emergency department (ED) and they result in approximately $12 billion in costs in the United States [1,2]. Recent epidemiologic study in Korea also showed that above 11,000 adult patients were diagnosed acute myocardial infarction (AMI) during 3 years [3]. On arrival in the ED, patients with acute chest pain typically undergo electrocardiograms (EKG) and cardiac enzyme determination to rule-in or rule-out acute coronary syndrome (ACS) [4,5]. ST-segment elevation/depression or elevated cardiac enzymes can be recognized for needed urgent coronary intervention [6,7]. If these tests do not yield a diagnosis of ACS, the patient generally undergoes repeat testing or admitted to the observational unit [8].

Coronary computed tomographic angiography (CCTA) is a non-invasive method to evaluate the coronary arteries [9,10]. Recent emphasis has been on the application of CCTA in the triage of low-risk chest pain patients which greater likelihood of discharge directly from the ED and reduces length of stay and time to discharge, when compared with the current standard of care [11]. Current guidelines recommended CCTA is an alternative to stress imaging techniques for ruling out stable ischemic heart disease (class IIa, LOE C) and subjects with a non-conclusive exercise EKG or stress imaging test or who have contraindications to stress testing (class IIa, LOE C) [12].

Despite improvement in the diagnostic care of chest pain patients presenting ED, the rule-out of ACS in patients without ST changes in EKG and without elevation of cardiac biomarkers (troponin or creatine-kinase myocardial band) is still challenging [12,13]. Routinely performing CCTA can help physicians to decide in this cohort; however, CCTA has a radiation exposure, the need for a contrast medium that is potentially allergenic and nephrotoxic, and the need for rate control [14,15]. Thus, CCTA is not recommended as a screening test in an asymptomatic individual without clinical suspicion of ACS (class III) [8]. Despite this, ambiguous cases are common in the clinical field and the potential use of CCTA in chest pain patients without ST changes and troponin elevation in the ED setting is promising. Therefore, our purpose was to develop a prediction model for finding patients with a significant stenosis in CCTA to suggest the potential candidate for CCTA in chest pain patients without ST changes and troponin elevation in ED.

## 2. Materials and Methods

We performed a retrospective, observational, cohort study between January 2017 and December 2018 at ED, which has an annual volume of approximately 130,000 patients in tertiary referral center in Seoul, Korea. All adult patients with acute chest pain those who were examined CCTA for ruling out ACS were reviewed. Exclusion criteria were those who had any ischemic EKG patterns, had been conducted CCTA other than assessing for ACS, and had elevated cardiac enzymes. Furthermore, we also precluded patients with previous history of coronary artery bypass graft surgery and non-determined results on report of CCTA because both cases were hard to determine and classify the culprit arteries. The study protocol was approved by the institutional review board of Asan Medical Center (No. 2019–0452), which waived the requirement for informed consent due to the retrospective nature of this study.

The data were extracted from electrical health records, included demographics, initial vital signs, underlying illnesses, cardiovascular risk factors, and associated symptoms. The primary physicians on duty described the pain characteristics on medical records and the co-authors of this study (J.-s.K., and Y.S.S) classified the typical and atypical chest pain based on Diamond-Forrester criteria [16,17]. In brief, typical chest pain included sensations of pressure, squeezing, heaviness, weight, vise-like aching, burning, tightness. Atypical chest pain included pleuritic, sharp, pricking, knife-like, pulsating, lancinating, chocking characteristics [16,17]. Blood samples were performed within 10 min, and troponin I (TnI) levels had been checked in all study population. A three-site sandwich immunoassay using troponin-I ultra-direct chemiluminometric technology (Siemens Healthineers, Erlangen, Germany) was used for measuring the troponin levels with a value of 0.04 ng/mL, which represents the 99th percentile reported in the normal population [18]. Because of the retrospective design, the repetition of enzyme follow-ups was different for each patient and elevated enzyme levels were evaluated by using all of TnI levels during ED stay. Ischemic EKG patterns were included ST-segment elevations (≥1 mm), ST-segment depression (≥1 mm), or primary T wave inversion (≥1 mm) on consecutive leads. The interpretations of the EKG were reviewed by two independent board-certified emergency medicine physicians (J.-s.K., Y.S.S., and S.A.), and when there were any debates, another reviewer (W.Y.K.) was invited for agreement.

Multidetector coronary CTA was performed using a dual source scanner (Somatom Definition, Siemens, Germany). Isosorbide dinitrate (Isoket spray; UCB Pharma, Monheim, Germany) were scattered into the patient’s oral cavity before contrast injection. A 50–70 mL bolus of iodinated contrast (Iomeprol; Bracco, Milan, Italy) was administrated at a rate of 4.0 mL/s, followed by saline. The tube voltage and tube current-time product were adjusted in accordance with each patients’ body sizes. All CCTAs were reported by experienced board-certified cardiovascular radiologists on duty and reviewed by the emergency medicine physicians (J.-s.K., Y.S.S., and S.A.) with blinding patients’ information. The primary endpoint was the presence of significant stenosis on CCTA, defined as narrowing above 70% diameter in cross-section at any main branches, including right, left anterior descending, left circumflex arteries, diagonal, and obtuse marginal artery.

All statistical analyses were performed using IBM SPSS Statistics V23.0 (SPSS Inc., Chicago, IL, USA). We classified the study population into two groups (i.e., not significant vs. significant group). Categorical variables were presented as counts with percentages and analyzed using the Chi-square test. Continuous variables were expressed as median with interquartile range (IQR) and using the Mann-Whitney U test because of a non-normal distribution of data. In the model development cohort, univariate analyses were conducted to explore the association between various risk factors and the occurrence of significant stenosis on CCTA. Furthermore, the logistic multivariate regression models were conducted to finding independent risk factors with statistically different parameters between groups in univariate analysis (*p* value < 0.1). A risk scoring model was created with the five variables associated with severe stenosis in results of CCTA: age 65 years or older, male, typical chest pain, diabetes, and previous history of ACS. We assigned each point based on the adjusted odds ratio (OR). The Hosmer-Lemeshow test was conducted to check the fitness of the model (i.e., *p* > 0.05). Receiver operating characteristic (ROC) curve with the area under the curve (AUC) of the prediction model were conducted to evaluate the discriminatory power of new model. Calculating cut-off values were used by the Youden’s index (sensitivity + specificity − 1). Sensitivity and specificity were computed through standard statistical method. A *p*-value of < 0.05 was considered significant.

## 3. Results

During the study period, 1247 patients who performed CCTA for evaluating ACS in ED were eligible (Figure 1). We excluded 343 patients for following reasons: (1) 283 had elevated TnI; (2) 28 had ischemic ST-wave changes initial electrocardiography; (3) 31 patients with previous coronary artery bypass graft; and (4) 1 patient had non-visualization of coronary artery on radiologic report. Finally, remaining 904 patients were included. Among these, 134 patients (14.8%) were shown severe stenosis on any of coronary arteries. All of the patients with significant stenosis were conducted coronary angiography and 125 (93.3%) had confirmed significant stenosis on angiography. Moreover, some patients without stenosis (*n* = 154, 20.0%) were also performed coronary angiography based on decisions of the cardiovascular physicians on duty and only 2 patients (1.3%) were found significant stenosis on angiography (Appendix A).

### 3.1. Baseline Characteristics of the Study Population

The baseline characteristic of the study population is presented in Table 1. Median age was 61.0 years with male predominant in both groups. Significant stenosis group showed older age (66.0 vs. 61.0 years, *p* < 0.001) and more male gender (79.9 vs. 56.6%, *p* < 0.001) than that of not significant group. All initial vital signs were similar between two populations. Regarding to underlying illnesses, hypertension (52.2 vs. 40.6%, *p* = 0.012), diabetes (28.4 vs. 15.8%, *p* < 0.001), obesity (3.0 vs. 0.5%, *p* = 0.005), and previous history of AMI (55.2 vs. 19.2%, *p* < 0.001) were more frequent in patients with severe stenosis. Moreover, patients with significant narrowing coronary artery had more common sensation of typical chest pain and less atypical characteristics than that of patients without significant stenosis. Among laboratory results, severe stenosis group showed lower platelet (218.0 vs. 233.0 × 10^3^/µL, *p* = 0.002), higher D-dimer (0.33 vs. 0.29 µg/mL, *p* = 0.023), TnI (0.006 vs. 0.006 ng/mL, *p* = 0.001), and creatinine levels (0.91 vs. 0.83 mg/dL, *p* < 0.001) than that of not severe group.

### 3.2. New Prediction Model for the Occurrence of Significant Stenosis on CCTA

Table 2 showed the logistic regression multivariate analysis and the new predicting model for the occurrence of significant stenosis on CCTA. Age above 65 years (adjusted OR 1.914 [95% CI 1.265–2.897], 2 points), male (adjusted OR 2.940 [95% CI 1.809–4.780], 3 points), typical chest pain (adjusted OR 2.186 [95% CI 1.446–3.304], 2 points), diabetes (adjusted OR 1.838 [95% CI 1.155–2.926], 2 points), and history of ACS (adjusted OR 3.442 [95%CI 2.283–5.190], 3 points) were revealed as independent risk factors in multivariate model. Based on these, we developed the new prognostic model for predicting significant stenosis on CCTA, and the Hosmer-Lemeshow test showed that the model fitted the data well (*p* = 0.848).

### 3.3. Diagnostic Performance of the New Prediction Model

Figure 2 shoed observed occurrence of significant stenosis according to the new score. Higher score tent to had higher proportion of the severe narrowing on coronary arteries. Figure 3 were ROC curve of the new model, and AUC was 0.782 (95% CI 0.742–0.822), *p* < 0.001. Cut-off value of 5 point which was calculated by using the Youden index showed 86.6% of sensitivity and 56.4% of specificity (Table 3).

## 4. Discussion

This study was to develop a new, simple scoring model with numerous clinical variables in identifying patients with acute chest pain who might be eligible for requiring CCTA in ED. We found five independent parameters and confirmed good prediction power for predicting severe stenosis on CCTA.

Applying rule-in or rule-out algorithms with clinical assessment, 12- lead EKG, and troponin is effectively sort out the patients with high risk of ACS [19,20]. Meanwhile, patients without ischemic patterns of EKG and negative cardiac enzymes may need further evaluations, including stress imaging or anatomical assessment through CCTA [2]. Recent randomized controlled trials revealed that risk stratification with CCTA in ED could be safe and effective [1,21]. However, as CCTA utilization has increased in recent years, the overuse of unnecessary CCTA for patients with chest pain also has increased [22]. Our results showed that 14.8% of patients those who had not ischemic EKG and normal level of TnI had a significant stenosis in CCTA. This incidence of severe stenosis among patients with low to intermediate risk for ACS was similar with previous observational study (13%, 79/196) [23]. This relatively high occurrence rate might be due to including the patients conducting CCTA.

Risk stratification of patients with acute chest pain has been extensively developed in recent year [24]. Among various tools, the Thrombolysis in Myocardial Infarction (TIMI) score, the Global Registry of Acute Coronary Events (GRACE) score, and the History, Electrocardiogram, Age, Risk factors, and initial Troponin (HEART) score were widely accepted [24]. Because the TIMI and GRACE score were not specifically designed for patients in ED setting, their performance has been limited [25]. On the other hand, the HEART score which was developed for ED patients showed superior performance than that of others in numerous validation studies [26]. However, these risk stratifications were focused on the occurrence of major adverse cardiac events and did not provide any detail criteria for conducting additional tests, including CCTA. Risk model for deciding to perform CCTA is more intuitive and important to quickly and accurately identify high risk patients to optimally allocate ED resources. Moreover, CCTA is limited in various situations including patients with tachycardia or irregular heartbeat; not round-clock-availability for 24 h; contrast and radiation issues [27].

The latest 2019 European Society of Cardiology (ESC) guidelines for chronic coronary syndromes updated the assessment tool for patients with stable chest pain, called pre-test probability for obstructive CAD [8]. This score included age, sex, dyslipidemia, hypertension, angina typicality, diabetes mellitus, and smoking status. Furthermore, recent validation study proved that patients with a pre-test probability > 15% might be beneficial for conducting CCTA [28]. However, calculating exact percentage of this score was quite complicated and not suitable for bedside examination. We created a simple risk stratification of ED for deciding whether performing CCTA or not with five variables. Age, previous AMI, diabetes were well-known universal clinical markers of risk for ACS. Age was made as a dichotomous parameter (≥65 years) to simplify. Typical chest pain based on Diamond-Forrester criteria, such as a retrosternal perception of pressure or heaviness were also included. Previous history of acute myocardial infarction might be definitely high-risk factor for recurrent ACS. However, if patients with history of AMI suffered from different pain characteristics, weak pain degree, and had normal EKG and enzyme levels, routinely performing direct coronary angiography might be too invasive to manage patients. Our study showed that about half of patients did not have any coronary stenosis and unnecessary to conduct any additional interventions. In this manner, we added a variable of previous acute myocardial infarction among patients with normal EKG and cardiac enzyme levels. Cut-off value above 5 point showed similar observed prevalence of significant stenosis with that of the pre-test probabilities.

We noticed several limitations in our study. It was conducted within a single center and thus may not be applied in other ED practice environments. Because of the retrospective esign, the results should be interpreted with caution, because even though we use a standardized format for electronic medical records, not all patients were able to give exact information required for chest pain characteristics, which would reflect data abstraction errors. In case of unstable angina, defined as negative results of cardiac biomarkers and no ischemic EKG changes, young patients (≤65 years old) with atypical chest pain could have low score and could discharge without any further tests. However, the past research reported that age under 65 with atypical angina showed a quite low prevalence of significant stenosis on CCTA and coronary angiography [16]. Therefore, patients with unstable angina those who had age under 65, atypical, female, without diabetes, and history of AMI, might be a very small portion of the total population. Moreover, interpretations of EKG might be erroneous due to subtle ST-segment changes. However, the effect of this error would not be significant because four investigators reviewed all ambiguous cases and made decisions. Finally, we did not collect data of functional tests, such as stress echocardiography, radionuclide myocardial perfusion imaging, or single photon emission computed tomography, and did not confirm the true clinical impact of anatomical stenosis on CCTA.

In conclusion, among chest pain patients without ST changes and troponin elevation, the new score would be helpful to identify potential candidate for CATA such as patients with significant stenosis. Further validation studies are needed.

## Figures and Tables

**Figure 1 jcm-10-05442-f001:**
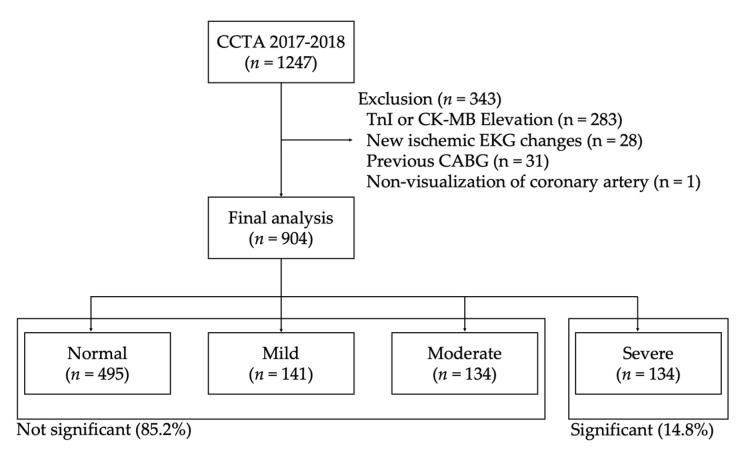
Study flowchart.

**Figure 2 jcm-10-05442-f002:**
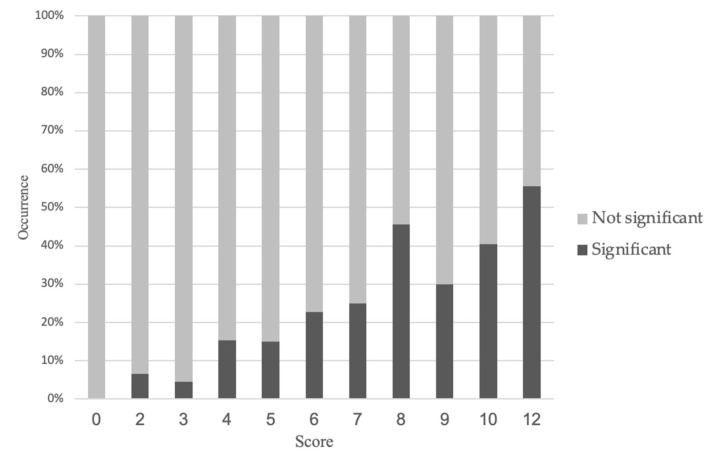
Observed occurrence of significant stenosis according to the new score.

**Figure 3 jcm-10-05442-f003:**
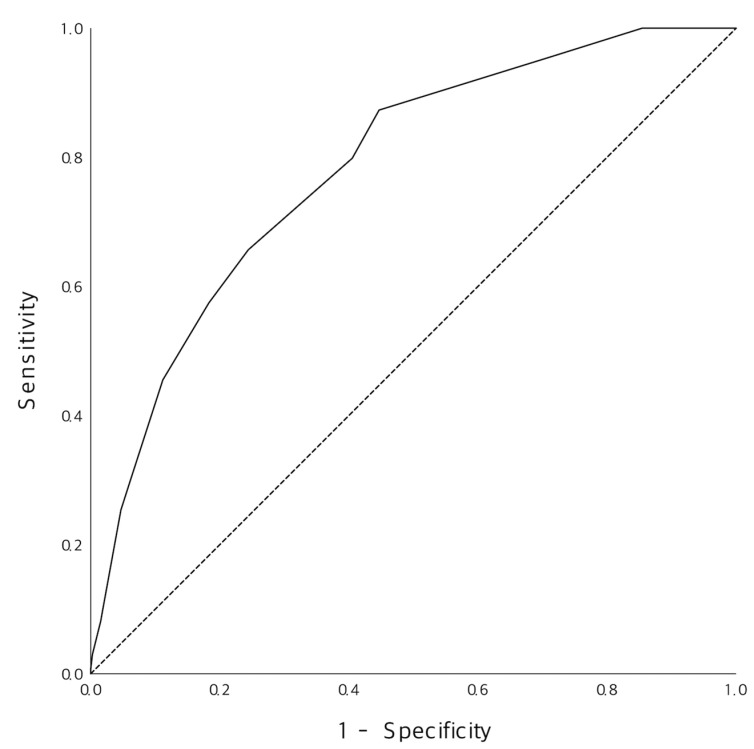
Comparison of the predictive ability of the new score with that of the development and validation set.

**Table 1 jcm-10-05442-t001:** Baseline characteristics of the study population.

Variables	Total (*n* = 904)	Not Significant (*n* = 770)	Significant (*n* = 134)	*p*-Value
Age, years	61.0 (54.0–70.0)	61.0 (52.0–70.0)	66.0 (58.0–80.0)	<0.001
Male	543 (60.1)	436 (56.6)	107 (79.9)	<0.001
Vital signs ^1^				
SBP, mmHg	141.0 (128.0–155.0)	141.0 (128.0–155.0)	142.0 (126.5–157.0)	0.852
DBP, mmHg	85.0 (76.0–94.0)	85.0 (77.0–94.0)	84.0 (73.5–92.5)	0.129
Pulse rate	78.0 (68.0–89.0)	78.0 (68.0–90.0)	76.0 (67.0–87.0)	0.202
Respiratory rate	18.0 (18.0–20.0)	18.0 (18.0–20.0)	18.0 (18.0–20.0)	0.992
Underlying disease				
Hypertension	383 (42.4)	313 (40.6)	70 (52.2)	0.012
Diabetes	160 (17.7)	122 (15.8)	38 (28.4)	<0.001
Hyperlipidemia	157 (17.4)	136 (17.7)	21 (15.7)	0.575
Current smoking	79 (8.7)	63 (8.2)	16 (11.9)	0.155
Obesity	8 (0.9)	4 (0.5)	4 (3.0)	0.005
History of AMI	222 (24.6)	148 (19.2)	74 (55.2)	<0.001
History of stroke	31 (3.4)	25 (3.2)	6 (4.5)	0.470
Family history of CAD	25 (2.8)	22 (2.9)	3 (2.2)	0.687
Symptoms				
Typical pain ^2^	243 (26.9)	178 (23.1)	65 (48.5)	<0.001
Atypical pain ^3^	438 (48.5)	385 (50.0)	53 (39.6)	0.026
Radiating	199 (22.0)	167 (21.7)	32 (23.9)	0.572
Diaphoresis	133 (14.7)	106 (13.8)	27 (20.1)	0.054
Laboratories				
Hb, mg/dL	13.8 (12.6–14.8)	13.9 (12.7–14.9)	13.7 (12.6–14.8)	0.292
Platelet, ×10^3^/µL	231.0 (197.0–275.0)	233.0 (199.0–278.0)	218.0 (180.0–259.5)	0.002
PT, INR	0.99 (0.96–1.04)	0.99 (0.96–1.04)	1.00 (0.96–1.06)	0.357
D-dimer, µg/mL	0.29 (0.19–0.50)	0.29 (0.19–0.49)	0.33 (0.21–0.60)	0.023
CK-MB, ng/mL	0.95 (0.50–1.80)	0.90 (0.40–1.70)	1.10 (0.60–2.00)	0.067
TnI, ng/mL	0.006 (0.006–0.006)	0.006 (0.006–0.006)	0.006 (0.006–0.008)	0.001
Creatinine, mg/dL	0.84 (0.71–0.97)	0.83 (0.70–0.96)	0.91 (0.77–1.05)	<0.001

^1^ Vital signs were checked on ED arrival. ^2^ Typical chest pain included sensations of pressure, squeezing, heaviness, weight, vise-like aching, burning, tightness. ^3^ Atypical chest pain included pleuritic, sharp, pricking, knife-like, pulsating, lancinating, chocking characteristics. Abbreviations: SBP, systolic blood pressure; DBP, diastolic blood pressure; AMI, acute myocardial infarction; CAD, coronary artery disease; Hb, hemoglobin; PT, prothrombin time; INR, international normalized ratio; CK-MB, creatinine kinase-myocardial band; TnI, troponin I.

**Table 2 jcm-10-05442-t002:** Multivariate analysis and new prognostic model for predicting significant stenosis.

Variables	Multivariate Analysis			Points
	Adjusted OR	95% CI	*p*-Value	
Age ≥ 65 years ^1^	1.914	1.265–2.897	0.002	2
Male	2.940	1.809–4.780	<0.001	3
Typical chest pain	2.186	1.446–3.304	<0.001	2
Diabetes	1.838	1.155–2.926	0.010	2
History of AMI	3.442	2.283–5.190	<0.001	3
Sum				12

^1^ Cut-off value was calculated through Youden’s index method. Abbreviations: OR, odds ratio; CI, confidence interval; AMI, acute myocardial infarction.

**Table 3 jcm-10-05442-t003:** Diagnostic performance in predicting the significant stenosis in the validation set.

Cutoff	Sensitivity	Specificity	PLR	NLR	PPV	NPV
≥2	100.0	15.5	1.2	0.0	17.1	100.0
≥3	100.0	15.5	1.2	0.0	17.1	100.0
≥4	93.3	31.8	1.4	0.2	19.2	96.5
≥5	86.6	56.4	2.0	0.2	25.7	96.0
≥6	82.1	60.7	2.1	0.3	26.6	95.1
≥7	64.9	77.5	2.9	0.5	33.5	92.7
≥8	57.5	82.0	3.2	0.5	35.7	91.7
≥9	41.0	90.5	4.3	0.7	43.0	89.8
≥10	21.6	94.6	4.0	0.8	40.9	87.4
≥11	19.4	95.5	4.3	0.8	42.6	87.2
≥12	3.7	99.5	7.2	1.0	55.6	85.3

Abbreviations: PLR, positive likelihood ratio; NLR, negative likelihood ratio; PPV, positive predictive value; NPV, negative predictive value.

## Data Availability

Data sharing is not applicable to this article.

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
