# Peer review of "Use of Coronary CT Angiography to Predict Obstructive Lesions in Patients with Chest Pain without Enzyme and ST-Segment Elevation"

_jcm, 2021, doi:10.3390/jcm10225442_

Round 1
Reviewer 1 Report
I commend the authors on aiming for an interesting study, although there are several areas of concern:
Introduction: The introduction is confusing. On one hand, the authors report the apparent poor utility of CCTA, especially in this small population of patients who present with cardiac chest pain but with absence of EKG or enzyme changes. They then mention that CCTA is significant in this population?
Materials/Methods:
- Why were CABG patients excluded?
- Were patients with AKI or CKD included or excluded?
- Which guidelines did you use to define "typical" vs "atypical" chest pain?
- Were serial troponins performed or just one troponin?
- Were the ED physicians reviewing the EKG blinded to the patient ID? Were the CCTA readers blinded?
- In the abstract you report primary endpoint as a >70% narrowing but in the materials/methods you report it as >50%?
Results:
3.1.:
- Please be consistent in sighting p-values when reporting significant findings between baseline characteristics since you have reported p-values.
3.2:
- How did you determine the points system?
- You state in line 145-147 that the new prognostic model fitted the data well but P-value was non-significant. What does this mean?
Author Response
I commend the authors on aiming for an interesting study, although there are several areas of concern:
Introduction: The introduction is confusing. On one hand, the authors report the apparent poor utility of CCTA, especially in this small population of patients who present with cardiac chest pain but with absence of EKG or enzyme changes. They then mention that CCTA is significant in this population?
Response> We agreed with the reviewer’s opinion that the introduction was not logical and some words made readers confused. We changed the sentences to clarify the meaning as below.
“Despite improvement in the diagnostic care of chest pain patients presenting ED, the rule-out of ACS in patients without ST changes in EKG and without elevation of cardiac biomarkers (troponin or creatine-kinase myocardial band) is still challenging [12,13]. Routinely performing CCTA can help physicians to decide in this cohort; however, CCTA has a radiation exposure, the need for a contrast medium that is potentially allergenic and nephrotoxic, and the need for rate control [14,15]. Thus, CCTA is not recommended as a screening test in an asymptomatic individual without clinical suspicion of ACS (class III) [8]. Despite this, ambiguous cases are common in the clinical field and the potential use of CCTA in chest pain patients without ST changes and troponin elevation in the ED setting is promising. Therefore, our purpose was to develop a prediction model for finding patients with significant stenosis in CCTA to suggest the potential candidate for CCTA in chest pain patients without ST changes and troponin elevation in ED.” (page 2, line 47-58)
Materials/Methods:
Why were CABG patients excluded?
Response> The formal readings of CACTs for those who had CABG could be confusing because of abnormal anatomy. Therefore, it was difficult to determine and classify the culprit lesions of significant stenosis. We added the sentence in the Materials/Methods section.
“Furthermore, we also precluded patients with the previous history of coronary artery bypass graft surgery and non-determined results on the report of CCTA because both cases were hard to determine and classify the culprit arteries.” (page 2, line 65-67)
Were patients with AKI or CKD included or excluded?
Response> This retrospective, observational, cohort study did not exclude AKI or CKD. However, in generally we did not perform CACT in AKI or CKD in our ED. In this manner, median creatinine levels of the total population is 0.84 (Table 1) and the median level is 0.84, which means most patients do not suffer from AKI or CKD.
Which guidelines did you use to define "typical" vs "atypical" chest pain?
Response> We also agreed with the reviewer’s opinion that the classification of typical or atypical pain could be arbitrary and different according to gender. We followed the standard definitions for typical and atypical chest pain based on previous studies (i.e., Diamond-Forrester criteria). The primary physicians on the duty of the study facility described the pain characteristics on electronic medical records and the co-authors of this study classified the typical and atypical chest pain. We added references in the manuscript.
“The primary physicians on duty described the pain characteristics on medical records and the co-authors of this study (J.-s.K.., and Y.S.S.) classified the typical and atypical chest pain based on Diamond-Forrester criteria [16,17]. In brief, typical chest pain included sensations of pressure, squeezing, heaviness, weight, vise-like aching, burning, tightness. Atypical chest pain included pleuritic, sharp, pricking, knife-like, pulsating, lancinating, chocking characteristics [16,17].” (page 2, line 73-78)
Were serial troponins performed or just one troponin?
Response> Despite of the, we did not gather the protocoled data. If the patients had one troponin test, we used only one laboratory result. Most of the patients had serial troponin results within 3 hours because ED physicians in the study facility routinely performed follow-up biomarkers. However the limitations from retrospective design, we did not gather the protocolized data (exact same time point). Thus, we defined the elevated enzyme levels by using all troponin levels during ED stay in this study. For clarifying this issue, we added the sentence in the manuscript.
“Because of the retrospective design, the repetition of enzyme follow-ups was different for each patient and elevated enzyme levels were evaluated by using all of TnI levels during ED stay.” (page 2, line 83-85)
Were the ED physicians reviewing the EKG blinded to the patient ID? Were the CCTA readers blinded?
Response> All EKG findings were reviewed by the co-authors with blinded patients’ IDs. All CCTA findings were initially reported by trained cardiovascular radiologists on duty and reviewed by the co-authors with blinding IDs. We provided this detailed information in the manuscript.
“The interpretations of the EKG were reviewed by two independent board-certified emergency medicine physicians with blinding patients’ IDs (KJS, SYS, and AS), and when there were any debates, another reviewer (KWY) was invited for agreement.” (page 2, line 87-89)
“All CCTAs were reported by experienced board-certified cardiovascular radiologists on duty and reviewed by the emergency medicine physicians (J.-s.K., Y.S.S., and S.A.) with blinding patients’ information.” (page 2, line 95-98)
In the abstract you report primary endpoint as a >70% narrowing but in the materials/methods you report it as >50%?
Response> We are sorry for your inconvenience and we corrected typos in the materials/methods section.
“The primary endpoint was the presence of significant stenosis on CCTA, defined as narrowing above 70% diameter in cross-section at any main branches, including right, left anterior descending, left circumflex arteries, diagonal, and obtuse marginal artery.” (page 3, line 98-101)
Results:
3.1.:
Please be consistent in sighting p-values when reporting significant findings between baseline characteristics since you have reported p-values.
Response> We cite the P values consistently in the manuscript in the result section.
“Significant stenosis group showed older age (66.0 vs. 61.0 years, P < 0.001) and more male gender (79.9 vs. 56.6%, P < 0.001) than that of not significant group. All initial vital signs were similar between two populations. Regarding to underlying illnesses, hypertension (52.2 vs. 40.6%, P = 0.012), diabetes (28.4 vs. 15.8%, P < 0.001), obesity (3.0 vs. 0.5%, P = 0.005), and previous history of ACS (55.2 vs. 19.2%, P < 0.001) were more frequent in patients with severe stenosis. Moreover, patients with significant narrowing coronary artery had more common sensation of typical chest pain and less atypical characteristics than that of patients without significant stenosis. Among laboratory results, severe stenosis group showed lower platelet (218.0 vs. 233.0 ×103/µL, P = 0.002), higher D-dimer (0.33 vs. 0.29 µg/mL, P = 0.023), TnI (0.006 vs. 0.006 ng/mL, P = 0.001), and creatinine levels (0.91 vs. 0.83 mg/dL, P < 0.001) than that of not severe group.” (page 4, line 136-147)
3.2:
How did you determine the points system?
Response> We determined the points based on the adjusted odds ratio because odds could reflect the potential to occur disease. For example, the variable of male had 2.940 of adjusted OR and we setpoint 3. We added the sentence in the method section
"We assigned each point based on the adjusted odds ratio (OR)." (page 2)
You state in lines 145-147 that the new prognostic model fitted the data well but P-value was non-significant. What does this mean?
Response> In the Hosmer-Lemeshow test, like most goodness of fit tests, a small P value (usually < 0.05) mean that the model is not a good fit. Even though this test does not exclude the problems of overfitting or low test power, a high P value commonly is considered a good fit for the model. We added the P value in the method section for clarifying our findings.
“The Hosmer-Lemeshow test was conducted to check the fitness of the model (i.e., P > 0.05).” (page 3, line 113-114)
Reviewer 2 Report
This is a useful and important study for assessment of patients at low risk for obstructive CAD. However, the indications for CCTA are flawed and patients with high pretest probability (that is history of ACS), may have been better suited for direct coronary angiography.
ACS also encompasses unstable angina, defined as negative cardiac biomarkers and no significant/ischemic EKG changes. If the pretest probability for obstructive disease is high enough, patients undergo invasive angiography. It is assumed that the patients did not have high pre-test probability in this study, but it has not been stated as such.
Use of CCTA in the Emergency Room is not uncommon as various clinical trials have already been completed showing its utility/benefits of excluding significant disease given strong negative predictive value.
This is a study that was conducted in Korea but in the introduction, discussion is about the United States.
There are standard definitions for typical and atypical chest pain. In the United States, the Diamond Forrester criteria are used. Are these descriptions as noted in the Methods section specific criteria in Korea or subjective classifications?
One of the univariate predictors, previous history of ACS, is very broad and there is no clarification of whether these patients had prior revascularization with stents. If so any recurrent presentation would put them in a higher pre-test probability group (as noted above) and may be the candidates where CCTA would not have been the best test. This is confirmed as it is the variable with the highest odds ratio.
The accuracy of CCTA reading was not confirmed by angiography, at least the data for this is not presented. Were all significant stenoses confirmed by angiography? What was the result of the nonsignificant stenoses patients – despite their symptoms, did no patients in this group undergo angiography?
Author Response
This is a useful and important study for assessment of patients at low risk for obstructive CAD. However, the indications for CCTA are flawed and patients with high pretest probability (that is history of ACS), may have been better suited for direct coronary angiography.
Response> Thanks for the generous comments. We totally agreed with the reviewer’s opinion that the previous history of ACS had a higher risk of recurrent ACS. It is more economic and safer for patients who have similar or severe pain to conduct direct coronary angiography as soon as possible. However, if patients with a history of ACS suffered from different pain characteristics, weak pain degree, and had normal EKG and enzyme levels, routinely performing direct coronary angiography might be too invasive to manage patients. Our study showed that about half of the patients did not have any coronary stenosis and unnecessary to conduct any additional interventions. Moreover, medical resources, including intensive care units, general wards, and emergent direct coronary angiography, might be not always available in ED. In this manner, we tried to develop a new scoring system for helping physicians to decide whether further exams could be helpful or not. We added a sentence in the discussion section to help clarify our intentions as below.
“Previous history of acute myocardial infarction might be definitely a high-risk factor for recurrent ACS. However, if patients with a history of ACS suffered from different pain characteristics, weak pain degree, and had normal EKG and enzyme levels, routinely performing direct coronary angiography might be too invasive to manage patients. Our study showed that about half of the patients did not have any coronary stenosis and unnecessary to conduct any additional interventions. In this manner, we added a variable of previous acute myocardial infarction among patients with normal EKG and cardiac enzyme levels.” (page 7, line 224-231)
ACS also encompasses unstable angina, defined as negative cardiac biomarkers and no significant/ischemic EKG changes. If the pretest probability for obstructive disease is high enough, patients undergo invasive angiography. It is assumed that the patients did not have high pre-test probability in this study, but it has not been stated as such.
Response> We totally agreed with the reviewer’s opinion that ACS included unstable angina which had negative cardiac enzyme and no EKG changes. Furthermore, if patients with unstable angina suffered from an atypical type of chest pain, our scoring system did not catch out them safely because new scoring did not include onset/duration of symptom, and escalating pattern which was the clue of unstable angina. That might be the reason why the cut-off value above 5 had 96.0 of negative predictive value. Meanwhile, the past article reported that age under 65 with atypical angina showed a quite low prevalence of significant stenosis on CCTA (Bing, R. et al. Our Heart J, 2020, 6, 293-300). Therefore, patients with unstable angina those who had age under 65, atypical, female, without diabetes, and history of AMI, might be a very small portion of the total population.
“In case of unstable angina, defined as negative results of cardiac biomarkers and no ischemic EKG changes, young patients (≤ 65 years old) with atypical chest pain could have low score and could discharge without any further tests. However, the past research reported that age under 65 with atypical angina showed a quite low prevalence of significant stenosis on CCTA and coronary angiography [16]. Therefore, patients with unstable angina those who had age under 65, atypical, female, without diabetes, and history of AMI, might be a very small portion of the total population.” (page 7, line 238-244)
Use of CCTA in the Emergency Room is not uncommon as various clinical trials have already been completed showing its utility/benefits of excluding significant disease given strong negative predictive value.
Response> We also agreed with the reviewer’s opinion that the previous trials proved the usefulness of CCTA in the ED for rule out a significant coronary artery disease. However it may not be practical and unnecessary to do CCTA in all chest pain patients without cardiac enzyme elevation (limited medical resources, radiation/radiocontrast issues, and cost). In our study found some clue that will be helpful to identify potential candidate for CCTA. We added this detail in the discussion section as below.
“However, as CCTA utilization has increased in recent years, the overuse of unnecessary CCTA for patients with chest pain also has increased [22].” (page 6, line 192-193)
This is a study that was conducted in Korea but in the introduction, discussion is about the United States.
Response> Thanks for the great suggestion. We added the recent epidemiological study conducted in Korea in the introduction and discussion section.
“Recent epidemiologic study in Korea also showed that above 11,000 adult patients were diagnosed acute myocardial infarction during 3 years [3].” (page 1, line 32-33)
“However, as CCTA utilization has increased in recent years, the overuse of unnecessary CCTA for patients with chest pain also has increased [22].” (page 6, line 192-193)
There are standard definitions for typical and atypical chest pain. In the United States, the Diamond Forrester criteria are used. Are these descriptions as noted in the Methods section specific criteria in Korea or subjective classifications?
Response> We also followed the standard definitions used in previous studies which were classified based on the Diamond Forrester criteria. We changed the fundamental reference to clear these criteria. The primary physicians on the duty of the study facility described the pain characteristics on electronic medical records and the co-authors of this study classified the typical and atypical chest pain.
“The primary physicians on duty described the pain characteristics on medical records and the co-authors of this study (J.-s.K., and Y.S.S) classified the typical and atypical chest pain based on Diamond-Forrester criteria [16,17]. In brief, typical chest pain included sensations of pressure, squeezing, heaviness, weight, vise-like aching, burning, tightness. Atypical chest pain included pleuritic, sharp, pricking, knife-like, pulsating, lancinating, chocking characteristics [16,17].” (page 2, line 73-75)
One of the univariate predictors, previous history of ACS, is very broad and there is no clarification of whether these patients had prior revascularization with stents. If so any recurrent presentation would put them in a higher pre-test probability group (as noted above) and may be the candidates where CCTA would not have been the best test. This is confirmed as it is the variable with the highest odds ratio.
Response> We agreed with the reviewer’s opinion that previous history of ACS is very broad and patients with prior revascularization with stents will be suitable for CAG. The definition of previous history of ACS was the patients who had previous history of unstable angina or Non-STEMI or STEMI. We added this definition in the method section. Although “prior revascularization with stents” will be more useful, this is not routinely evaluated at ED. Because of the limitation of retrospective study we cannot find data of “prior revascularization with stents”.
It is a big challenging for ED physicians that every patient with prior history of ACS should be underwent invasive coronary angiography or CACT or short-term observation if they do not have severe, typical chest pain and initial EKG and enzyme levels are normal. In our data showed 33% (74/222) patients with previous history of ACS had a significant lesion in CACT (Table 1). However our new prediction model (age, male, diabetes, history of ACS, and typical chest pain) was more accurate.
The accuracy of CCTA reading was not confirmed by angiography, at least the data for this is not presented. Were all significant stenoses confirmed by angiography? What was the result of the nonsignificant stenoses patients – despite their symptoms, did no patients in this group undergo angiography?
Response> The primary purpose of this study was to provide a simple score system to help the physician decide whether further image workup is necessary or not. However, we also agreed with the reviewer’s opinion that the significant lesion on CCTA should be confirmed by angiography. All of 134 patients with significant stenosis on CCTA conducted coronary angiography and 125 (93.3%) patients had confirmed significant stenosis on angiography. Among patients without significant stenosis (n = 770), 154 (20.0%) patients were performed coronary angiography based on decisions of the cardiovascular physicians on duty and only 2 patients (1.3%) were found significant stenosis on angiography. We added this detail with a supplementary table and the discussion section.
“All of the patients with significant stenosis were conducted coronary angiography and 125 (93.3%) had confirmed significant stenosis on angiography. Moreover, some patients without stenosis (n = 154, 20.0%) were also performed coronary angiography based on decisions of the cardiovascular physicians on duty and only 2 patients (1.3%) were found significant stenosis on angiography (Figure S1).” (page 3, line 126-131)
Round 2
Reviewer 1 Report
Thank you for your corrections.